# Effects of the Ketogenic Diet on Microbiota Composition and Short-Chain Fatty Acids in Women with Overweight/Obesity

**DOI:** 10.3390/nu16244374

**Published:** 2024-12-19

**Authors:** Müge Güzey Akansel, Murat Baş, Cansu Gençalp, Meryem Kahrıman, Eray Şahin, Hakan Öztürk, Gürsel Gür, Ceren Gür

**Affiliations:** 1Department of Nutrition and Dietetics, Institute of Health Sciences, Acibadem Mehmet Ali Aydinlar University, Istanbul 34752, Turkey; mugeguzey@gmail.com; 2Department of Nutrition and Dietetics, Faculty of Health Sciences, Acibadem Mehmet Ali Aydinlar University, Istanbul 34752, Turkey; murat.bas@acibadem.edu.tr (M.B.); meryem.kahriman@acibadem.edu.tr (M.K.); 3Biostatistics and Bioinformatics PhD Program, Institute of Health Sciences, Acibadem Mehmet Ali Aydinlar University, Istanbul 34572, Turkey; sahin.eray89@gmail.com; 4Department of Physiology, Faculty of Veterinary Medicine, Ankara University, Ankara 06110, Turkey; ozturkh@veterinary.ankara.edu.tr; 5Agriculture and Rural Development Support Institution, Ankara 06490, Turkey; gurselgur1974@gmail.com; 6Bağcılar Training and Research Hospital, University of Health Sciences, Istanbul 34200, Turkey; cerencalti@yahoo.com

**Keywords:** ketogenic diet, gut microbiota, short-chain fatty acids, cardiometabolic risk, dietary intervention

## Abstract

**Background/Objectives**: The ketogenic diet (KD) is a dietary model that can impact metabolic health and microbiota and has been widely discussed in recent years. This study aimed to evaluate the effects of a 6-week KD on biochemical parameters, gut microbiota, and fecal short-chain fatty acids (SCFAs) in women with overweight/obesity. **Methods**: Overall, 15 women aged 26–46 years were included in this study. Blood samples, fecal samples, and anthropometric measurements were evaluated at the beginning and end of this study. **Results:** After KD, the mean body mass index decreased from 29.81 ± 4.74 to 27.12 ± 4.23 kg/m^2^, and all decreases in anthropometric measurements were significant (*p* < 0.05). Fasting glucose, insulin, homeostasis model assessment of insulin resistance, hemoglobin A1C, urea, and creatinine levels decreased, whereas uric acid levels increased (*p* < 0.05). Furthermore, increased serum zonulin levels were noted (*p* = 0.001), whereas fecal butyrate, propionate, acetate, and total SCFA levels decreased (*p* < 0.05). When the changes in microbiota composition were examined, a decrease in beta diversity (*p* = 0.001) was observed. After the intervention, a statistically significant increase was noted in the Firmicutes/Bacteroidetes ratio (*p* = 0.001). Although Oscilibacter, Blautia, and Akkermensia relative abundances increased, Prevotella relative abundance and Bifidobacter abundance, which were the dominant genera before the KD, decreased. Moreover, the abundance of some pathogenic genera, including Escherichia, Klebsilella, and Listeria, increased. **Conclusions**: In healthy individuals, KD may cause significant changes in microbial composition, leading to dysbiosis and long-term adverse outcomes with changes in serum zonulin and fecal SCFA levels.

## 1. Introduction

The prevalence of obesity and metabolic diseases is rapidly increasing annually, and they are becoming one of the public health issues with a significant burden [1,2]. Therefore, health authorities or policy makers have implemented national or international strategies to prevent or mitigate these diseases [3,4]. Although the rate of increase has leveled off in some countries, no country has yet reversed the obesity epidemic [4]. In this direction, different alternatives have been attempted for diet, which is the cornerstone of obesity management, and new diet models have been directed [5,6,7]. The ketogenic diet (KD) is one of these alternatives. This diet, of which there are several types, aims to increase the proportion of fat while decreasing the proportion of carbohydrates to mimic the metabolic effects of starvation [8,9]. Mainly used as a key component of epilepsy treatment [10], the KD has been discussed in recent years for several diseases, including obesity [11], diabetes [12], cardiovascular diseases [13], and cancer [14].

Recently, microbiota has been frequently discussed in terms of maintaining metabolic health [15,16]. The concept of microbiota describes the microorganisms living in a specific area, including the mouth and gut [17]. The microbiota, which comprises various types of bacteria, archaea, fungi, protozoa, and viruses [16], functions as an essential organ in providing nutrition, regulating epithelial development, and directing innate immunity [18]. The diversity and species in the microbiota can be modulated by diet, drug use, physical activity and various lifestyle factors and that this homeostasis may play a role in metabolic disease development or prevention [15,16].

Furthermore, KD has been discussed as a dietary model that may be effective in modulating microbiota. Previous studies have reported that KD can alter microbial diversity and thus short-chain fatty acid (SCFA) production. Notably, these changes in the gut microbiota may be a communication pathway between the brain and the gut, potentially affecting the whole body [19,20,21]. Accordingly, it is believed that KD may also affect metabolic health in these mechanisms [19]. However, these mechanisms and short- and long-term effects remain unclear. Additionally, the effects of KD in women with overweight/obesity have been less investigated and have not yet been clarified. This study aimed to evaluate the effects of KD on metabolic health and gut microbiota composition in women with overweight/obesity. We hypothesized that gut microbial diversity will decrease following a KD. The primary outcomes were alpha (α) diversity expressed as the Chao1, observed, Shannon, and Invsimpson measurements and beta (β) diversity using Bray–Curtis dissimilarity scores.

## 2. Materials and Methods

### 2.1. Participants

Power analysis was performed according to the two-tailed *t*-test used for assessing the difference between two dependent means. The effect size was set as 0.80 based on Cohen’s large effect size, the alpha error probability was set as 0.05, and the power was set as 80%, resulting in a sample size of at least 15 participants.

Women aged 20–65 years with a body mass index (BMI) of >25 kg/m^2^ were included in this study. Pregnant and lactating women, patients with chronic diseases, those using antibiotics over the last 6 months, those using probiotic or symbiotic supplements over the last 6 months, and those with eating disorders were excluded. After providing general information about the study, all participants voluntarily agreed to participate and signed an informed consent form.

### 2.2. Study Design

This clinical study was conducted at a diet counseling center in Istanbul, Turkiye from October 2019 to December 2019. Before and after the KD, anthropometric measurements were taken, body composition was analyzed, and blood and fecal samples were collected.

The study protocol was approved by Acıbadem University Faculty of Medicine Research Evaluation Board (2019-7/18), and this study was conducted following the principles of the Declaration of Helsinki.

### 2.3. Dietary Intervention

The energy of the KD was calculated by reducing 500 kcal from the total energy expenditure. The KD contained <20 g of carbohydrate and >70% of energy from fat. The protein requirement was planned to be 1 g/kg. Participants were asked to consume fish 2 days a week, and the extra fat required for a KD was supplemented with olive oil. In the planned KD, vegetables, including artichokes, potatoes, carrots, beets, and tomatoes, and fruits were not included, however, green leafy vegetables and oil seeds, such as walnuts, almonds, and hazelnuts, were included to make the KD similar to the Mediterranean KD. A sample menu of the KD is presented in Appendix A. The KD was started after collecting the first blood and fecal samples. Participants were asked not to use nutritional supplements and not to change their physical activity levels during the study period. Participants’ compliance with the diet was monitored by daily messaging, and their ketosis status was evaluated by weekly urine ketone measurement (Ketostix Ames 2880, BAYER, Leverkusen, Almanya).

### 2.4. Food Intake Assessment

Three-day food records (2 days on weekdays and 1 day on weekends) were used for assessing baseline food intake. To verify the amounts consumed, the Food and Nutrition Photo Catalog was used. To calculate the intake of energy, macronutrients, and micronutrients, BeBIS version 7.2 was used.

### 2.5. Anthropometric Measurements and Body Composition

Body weight (BW), height, waist circumference (WC), hip circumference (HC), fat mass (FM), and fat-free mass (FFM) were measured; subsequently, BMI, waist-to-hip ratio (WHR), and waist-to-height ratio (WHtR) were calculated at baseline and after the study. BW was measured in the morning on an empty stomach, wearing thin clothes and without shoes, and using a digital scale with a sensitivity of 100 g. Height was measured using a stadiometer without shoes. The following formula was used for calculating BMI: weight (kg)/height^2^ (m^2^). WC was measured using an inflexible tape measure at the midpoint between the lowest rib and the cristae–iliac region, and HC was measured at the widest circumference from the side.

Body composition was evaluated using a bioimpedance analyzer, Tanita MC 780 MA (Tanita, Tokyo, Japan). Participants were informed in advance about the conditions required for BIA measurement.

### 2.6. Physical Activity Assessment

Participants were asked to record their daily physical activities in minutes according to the grouped physical activities. The time spent on grouped physical activities was ensured to be equal to 24 h in total. Activity levels of the participants were calculated in physical activity level (PAL).

### 2.7. Blood Sample Collection

Blood samples were collected at baseline and after the KD. The samples were collected in the morning following a 12-h fasting and were analyzed using routine methods in the Biochemistry Laboratory of Bagcılar Training and Research Hospital.

#### 2.7.1. Cardiometabolic Risk Determination

Serum fasting glucose, fasting insulin, hemoglobin A1C (HbA1C), total cholesterol (TC), triglyceride (TG), low-density lipoprotein cholesterol (LDL-C), high-density lipoprotein cholesterol (HDL-C), and C-reactive protein concentrations were used to determine the cardiometabolic risk. The homeostasis model assessment of insulin resistance (HOMA-IR) index was used as an indicator of insulin resistance and calculated as follows: HOMA-IR = fasting plasma insulin (µU mL^−1^) × fasting plasma glucose (FPG) (mmol L^−1^)/22.5.

#### 2.7.2. Liver and Renal Function Determination

Serum aspartate aminotransferase and alanine aminotransferase concentrations were used for evaluating liver function; serum urea and creatinine levels were used for evaluating renal function.

#### 2.7.3. Intestinal Permeability Determination

The serum zonulin-1 (ZO-1) concentration was used for determining the intestinal permeability.

### 2.8. Fecal Sample Collection

Fresh fecal samples were collected at baseline and after the KD. At each time point, fecal samples were collected from each participant in a sterile fecal collection tube and stored at −80 °C until further processing.

#### 2.8.1. Fecal DNA Extraction and 16S Amplicon Sequencing

All stages from sample processing to taxonomic annotation and abundance table preparation were completed by Epigenetiks Inc. (Istanbul, Turkiye). The ZymoBIOMICS DNA Miniprep Kit (Zymo Research, Irvine, CA, USA, Cat. No. D4300) was used for extracting DNA from up to 200 mg of each stool sample, and the Qubit dsDNA HS Assay Kit (Invitrogen, Waltham, Massachusetts, USA, Cat. No. Q32854) was used for quantifying the concentration of the extracted DNA. Following the manufacturer’s instructions (v.RAB_9053_V1_REVR_14AUG2019), library preparation was performed using a 16S barcoding kit (Oxford Nanopore Technologies, Oxford, UK, Cat. No. SQK-RAB204), including the entire 16S rRNA gene from regions V1–V9. Prepared libraries will be loaded into R9.4.1 FLO-MIN106D flow cells, and sequencing will be performed on a MinION sequencer (Oxford Nanopore Technologies).

Obtained raw FAST5 reads were converted to FASTQ format using Guppy (ver. 6.0.5). Primer sequences were removed from the amplicon reads, and quality trimming was performed using BBTools v.38.94 [22]. Reads passing quality filtering were clustered using Magicblast v.1.6.0 (Boratyn et al., 2019) by the Human Microbiome Project, using the custom-prepared reference of 16S rRNA gene regions of bacteria associated with human microbiome (16S NCBI reference sequences as of 12 August 2022). Consensus sequences were created, and .sam files were produced in Samtools [23]. Taxonomic annotations were determined using BLAST + v.2.12.0 [24] in the NCBI nr database (12 August 2022) using a sequence identity threshold of 95% for the genus level. Finally, relative abundance percentages were calculated at phylum and genus levels.

#### 2.8.2. Bioinformatic Analyses

All downstream data analysis steps were performed using R (v.4.1.3). To investigate the within-sample community diversity, alpha diversity analysis was performed using the “phyloseq” package (v.1.38.0). For that purpose, the observed features, Shannon index, and Simpson index were calculated at the phylogenic and genomic levels (10.1002/ece3.1155). The changes at baseline and after the KD were examined using the paired Wilcoxon signed rank test for each metric. Beta diversity analysis was performed by calculating Bray–Curtis dissimilarities using phylum- and genus-level bacterial relative abundances to evaluate the differences between sample diversity among the members of time groups. A permutational multivariate analysis of variance was performed between two-time groups using the “adonis” function of the “vegan” package.

Before proceeding to bacterial taxa comparisons between groups, filtering was applied to capture the bacteria present in most samples. Here, phyla with a ≥0.1% relative abundance and genera with a ≥0.05% relative abundance in at least 80% of the overall samples were retained, whereas the remainder was discarded. Consequently, 4 phyla and 10 genera abundances for 30 samples were obtained.

After obtaining the tables of operational taxonomic units (OTUs) by phylum and genus, the relative abundance was calculated for each OTU. To check whether these OTUs were normally distributed, the Shapiro–Wilk test was employed. As normal distribution was not obtained in the vast majority of cases, the non-parametric Mann–Whitney U test was employed for pairwise comparisons.

The Mann–Whitney U test was used for evaluating whether a statistically significant change existed between the density of each bacterium of the 15 study participants between the beginning of the study and after 6 weeks, and the *p* values obtained were corrected using the Benjamini–Hochberg (BH) method. After these tests, the changes in bacterial densities between the beginning of the study and after 6 weeks were separately calculated for each participant by taking the logarithm according to base 2, in case the high variation in the microbiota composition of the participants, the small number of participants, and the presence of a high number of bacteria at the genus level may overshadow the determination of species that may be significant with BH correction. For at least 10 of the 15 participants, bacteria that increased or decreased by a factor of two or more were identified and plotted.

In addition to differential abundance analysis, linear discriminant effect size (LEfSe) analysis was used for biomarker discovery on phylum and genus abundance tables. The analysis was conducted using the web-based Galaxy tool with the default settings.

#### 2.8.3. SCFA Determination

Two grams of fecal sample were transferred to a sterile spoon stool collection container. These samples were stored at −20 °C until fecal dry matter determination.

The night before the day of analysis, fecal samples were thawed from −20 °C to 4 °C. The samples were dried in an oven with perforated steel shelves at 65 °C for 48 h by completely removing the moisture and subsequently kept in a desiccator until reaching 20–25 °C. The samples were subsequently weighed on an analytical precision balance, and the amount of dry matter in the fecal samples was calculated by deducting the tare of the sample containers.

One gram of fecal sample was transferred to a sterile spoon stool collection container. For acidification, 5 mL of a 2-N HCl solution was added to these samples, and the vortex was centrifuged at 3000 rpm for 2–3 min to homogenize the sample with the solution. Homogenized samples were stored at −20 °C until the day of analysis.

SCFA analysis was performed in the Rumen Simulation Technique Laboratory of the Department of Physiology, Faculty of Veterinary Medicine, Ankara University. The night before the day of analysis, fecal samples were thawed from −20 °C to 4 °C and centrifuged at 15,000 rpm for 15 min using a refrigerated centrifuge at 4 °C. The supernatants were passed through nylon filters with a 0.2-µm pore diameter, placed in gas chromatography vials, and analyzed on an ACME-6100 gas chromatography system using an HP Innovax capillary column (30 m × 0.25 mm i.d. × 0.25-µm film thickness). Helium gas served as the carrier gas at a 1.8-mL/min flow rate. Injector and detector temperatures were set to 250 °C and 300 °C, respectively. The split ratio of the flame isonization detector was 1:40, and the oven program was set to 120 °C for 1 min, increasing by 10 °C/min to 265 °C and maintaining this temperature for 2 min. The injected sample was taken in 1 µL. Calibration was performed using 5, 10, 20, and 40 mmol/L standards of acetic, propionic, isobutyric, and butyric acids, respectively. To generate and evaluate the chromatograms, the Autochro-3000 software was used. SCFA results were calculated in µmol/gram of dry matter.

### 2.9. Statistical Analysis

The IIBM Statistical Package for the Social Sciences version 22.0 software (IBM Japan Ltd., Tokyo, Japan) was used for evaluating the findings obtained except for microbiota analysis. The Shapiro–Wilk test was used for assessing normality. For normally distributed continuous variables, descriptive statistics were expressed as means ± standard deviations, whereas for non-normally distributed variables, descriptions were presented as medians (25th and 75th percentiles). Pearson’s correlation analysis was used to examine the relationships between the parameters that conform to normal distribution, and Spearman’s correlation analysis was used to investigate the relationships between the parameters that do not conform to normal distribution. A *p* value of <0.05 (two-sided) was considered statistically significant.

## 3. Results

Overall, 15 women completed the study. Their mean age was 36.47 ± 6.45 years, and most of them (73.3%) were university graduates.

### 3.1. Changes in Energy and Macronutrient Intake After KD

During the KD, the mean energy intake was 1516.13 ± 177.18 kcal, of which carbohydrate, protein, and fat accounted for 5.4% ± 0.83%, 22% ± 1.41%, and 72.87% ± 1.77% of the daily energy intake. Compared with the baseline, participants’ carbohydrate and fiber intakes significantly decreased, whereas protein and fat intakes significantly increased after the study. No significant difference in energy intake was noted (Table 1).

### 3.2. Changes in Anthropometric Measurements After the KD

During the KD, participants lost 7.37 ± 2.16 kg of BW and 4.52 ± 1.8 of FM. A significant decrease was observed in weight, BMI, WC, HC, WHR, WHtR, and FM, whereas FFM and body water were significantly increased after the study (Table 2).

### 3.3. Changes in Biochemical Parameters After the KD

The biochemical parameters of the participants at baseline and after the KD are summarized in Table 3. Compared with the baseline, FPG, HbA1C, insulin, HOMA-IR, urea, and creatinine levels were significantly reduced after 6 weeks. In contrast, uric acid and serum ZO levels were significantly increased after the 6-week KD.

### 3.4. Changes in Fecal SCFAs After the KD

Compared with the baseline, fecal SCFA, acetate, propionate, and butyrate levels were significantly reduced after the study (Table 4). Furthermore, a significant moderate inverse correlation was observed between fecal SCFA and serum ZO levels after the study (*r* = 0.614; *p* = 0.015).

### 3.5. Evaluation of Alpha and Beta Diversity

The KD was associated with decreased alpha diversity metrics (Figure 1a), including observed index (*p* = 0.098), Chao1 index (*p* = 0.098), Shannon index (*p* = 0.187), and inverse Simpson index (*p* = 0.098). After the study, the intestinal microbiome of the participants were similar to each other with closer clustering (*p* = 0.001) (Figure 1b).

### 3.6. Evaluation of the Most Abundant Phyla

A total of 39 phyla were detected. The four phyla with the highest abundance at the beginning and after the study were Firmicutes (65.8% and 92.6%), Bacteroidetes (25.7% and 4.2%), Proteobacteria (1.2% and 1.4%), and Tenericutes (0.2% and 0.3%), respectively (Figure 2).

After the study, a statistically significant increase in the relative abundance of Firmicutes and a statistically significant decrease in the relative abundance of Bacteroidetes were observed. Before the KD, the Firmicutes-to-Bacteroidetes ratio was 5.94 ± 8.10, whereas that after the KD was 31.50 ± 36.56. The increase in the Firmicutes–Bacteroidetes rate was statistically significant (*p* = 0.001). Phyla with a two-fold or more variation are shown in Appendix A.

### 3.7. Evaluation of LefSe Analysis at the Phylum Level

Phyla with statistically significant differences in relative abundances between pre- and post-KD were identified as biomarkers using LefSe analysis. The criterion for the phylum to be significant is that the Lineer Discriminant Analysis (LDA) scores are higher than the specified threshold value of two. Although Actinobacteria, Thermotogae, Acidobacteria, Deinnococcus–Thermus, Deferribacteres, and Bacteroidetes were significantly more abundant before the study, and Firmicutes were significantly more abundant after the study (Figure 3).

### 3.8. Evaluation of the Most Abundant Genus

Overall, 2138 different genus were detected. The relative abundances of the ten genera with the highest abundance at the genus level are shown in Figure 4.

After the study, eight of the ten most abundant genera remained unchanged (Lachnospiracea incertae sedis [*p* > 0.05], Eubacterium [*p* > 0.05], Faecalibacterium [*p* > 0.05], Oscilibacter [*p* = 0.022], Bacteroides [*p* > 0.05], Rosebruia [*p* > 0.05], Ruminococcus [*p* > 0.05], and Clostridium), the abundance of Provetella (*p* = 0.025) and Coprococcus (*p* > 0.05) decreased, and the abundance of Lactobacillus (*p* > 0.05) and Blautian (*p* = 0.031) increased. Genera with a two-fold or more variation are shown in Appendix A.

### 3.9. Evaluation of LefSe Analysis at the Genus Level

The genus with statistically significant differences in relative abundances between pre- and post-KD were identified as biomarkers using LefSe analysis. The criterion for the genus to be significant is that the LDA scores are higher than the specified threshold value of two. In LefSe analysis, the most significant difference at the genus level before the study was in the Prevotella genus, whereas Oscilibacter showed the most significant difference after the study (Figure 5).

## 4. Discussion

The KD is steadily gaining popularity, especially owing to its effects on BW [25]. Previous studies have highlighted the effects of this dietary pattern on obesity, metabolic health, and several other health parameters, as well as on the microbiota [20,26,27]. We here evaluated the effects of a 6-week KD in a healthy population, considering the lack of data on healthy adults.

Evaluating the effects of the KD on anthropometric measurements revealed that BMI, body fat percentage, WHR, and WHtR significantly decreased, whereas lean mass percentage and muscle percentage significantly increased. Furthermore, we determined that the number of participants who were in the obese and severe obese categories before the intervention decreased after the intervention. Similar to our findings, previous studies conducted especially in obese populations have reported that the KD provides BW loss and an improvement in cardiometabolic risk factors [28,29]. Another study evaluating the effects of a KD without energy restriction for 6 weeks in healthy adults showed significant decreases in BW and body fat percentage after the KD [30]. Considering that WC, WHR [31], and WHtR [32] may be cardiometabolic risk indicators, these decreases in anthropometric measurements suggest that the KD can also be promising for healthy adults. However, these findings should be considered together with biochemical markers and it should not be forgotten that data on long-term effects are limited.

Metabolic risk markers are significant societal issues, considering their increasing prevalence and their associated complications, and the effects of the KD on this issue remain controversial [33]. In this regard, we also evaluated biochemical markers and showed that glucose, insulin, HOMA-IR, and HbA1C levels significantly decreased after the 6-week KD. Regarding this issue, a systematic review of studies evaluating the effects of a KD on glycemic control in patients with type 2 diabetes reported that it was effective in reducing fasting glucose and HbA1C levels [34]. Similarly, a systematic review of 20 studies investigating the effects of a KD on metabolic syndrome criteria reported that most of the included studies showed significant changes in glucose, insulin, HOMA-IR, and HbA1C levels after the intervention [33]. Another study examining the effects of a KD on a healthy population for 6 weeks reported decreased glucose and insulin levels after the intervention [30]. These findings, consistent with the literature, suggest that KD can positively affect glucose metabolism and insulin sensitivity. However, the idea that the KD may increase serum lipids due to its high-fat content is a matter of concern [35]. In the present study, a statistically significant change in TC, HDL-C, LDL-C, TG, LDL/HDL, and TC/HDL ratios after 6 weeks was not noted. This finding did not confirm that KD exerts an atherogenic effect on blood lipid levels.

The serum protein ZO is used as a peripheral marker for assessing intestinal permeability [36,37]. Microorganisms belonging to the intestinal microbiota, especially pathogenic species, cause increased release of ZO, which is associated with increased permeability in the ileum and jejunum [38]. We evaluated the effects of a 6-week KD on serum ZO levels and showed that serum ZO levels increased with the intervention (*p* = 0.001). Regarding this issue, Mörkl et al. [39] reported that individuals with higher serum ZO levels had higher energy and lipid consumption. In another study, Linsalata et al. [40] examined the effects of an 8-week very-low-calorie KD in participants with obesity and reported that the intervention had no significant effect on serum ZO levels. Consistent with these findings, the effects of diet and macronutrient composition on ZO levels remain controversial in the literature. However, while dietary fiber is generally accepted as a nutrient that protects intestinal integrity, lipids promote increased intestinal permeability in the literature [41]. Considering the increased lipid intake and decreased fiber intake with ketogenic diet in our study, we think that the change in macronutrient composition may have damaged the intestinal barrier integrity. In addition to all these, the increase in intestinal permeability shown by increased zonulin may also explain the increase in serum uric acid levels. This is because if the integrity of the intestine is compromised, uric acid levels may increase as a marker of the inflammatory process as lipopolysaccharide leaks into the bloodstream and activates innate immune mechanisms [42]. These findings point to an interaction between permeability and uric acid levels.

Dietary fiber fermentation is a significant function of the microbiota, and fermentation of this nutrient produces SCFAs. These metabolites may play a role in the development and treatment of gastrointestinal, metabolic, cardiovascular, and gut–brain disorders [43]. Moreover, the dietary composition can influence the production of these metabolites. Regarding this issue, Rad et al. [44], in their study evaluating the effects of an energy-restricted low-carbohydrate diet in women with obesity, reported that this diet increased fecal SCFA levels. Brinkworth et al. [45], in their study evaluating the effects of low-carbohydrate high-fat and high-carbohydrate low-fat diets for 8 weeks, noted that the low-carbohydrate diet exerted a negative effect by causing significant decreases in fecal acetate, butyrate, and total SCFA levels. These findings indicate that the effects of diet composition on fecal SCFA production remain unclear in the literature. In our study, we observed significant decreases in fecal butyrate, propionate, acetate, and total SCFA levels after the 6-week KD. In addition, participants’ dietary fiber intake was significantly reduced after the ketogenic diet intervention. This level is below the Institute of Medicine’s adequate intake recommendations of 38 g/day (for men) and 25 g/day (for women) [46]. This finding suggests that KD can negatively affect healthy individuals. We believe that this situation may occur because of the low carbohydrate and fiber contents of KD, considering that SCFA production is mainly due to dietary fiber [43].

Dietary nutrient content is a significant component affecting microbial diversity, and a high dietary fat intake can reduce microbiota richness and diversity [47]. In studies on KD, although taxonomic differences were observed in the microbiota after the diet, different results were obtained in alpha and beta diversity. Although some studies have reported no changes in alpha and beta diversity [48,49], in some studies it was observed that beta [50] and alpha diversity [51] decreased. In the present study, we evaluated alpha and beta diversity using different indices. Although a tendency to decrease in alpha diversity was observed with the change in taxonomic diversity, we did not find this decrease statistically significant. The change in beta diversity after the 6-week KD indicated that the variations in the microbiota of individuals decreased and exhibited closer clustering, and this decrease in beta diversity was statistically significant.

KD, which can affect fecal SCFA production, alpha, and beta diversity, can also change the levels of some phyla and genera in the intestinal microbiota [52]. In this regard, when we examined the effects at the phylum level, we showed that although the abundances of Actinobacteria, Thermotogae, Acidobacteria, Deinnococcus–Thermus, Deferribacteres, and Bacteroidetes were significantly evident before the intervention, the abundance of Firmicutes was significantly evident after the intervention. Similarly, Wang et al. [53], in their study wherein they administered a high-fat diet intervention to mice for 12 weeks, reported that the abundances of Firmicutes and Proteobacteria increased, whereas those of Bacteroidetes and Actinobacteria decreased. In another study conducted by Ma et al. [54], wherein a KD was administered to healthy mice for 16 weeks, Firmicutes, Actinobacteria, and Verrucomicrobia phyla significantly increased, whereas Proteobacteria decreased after the intervention. Although there are some differences between studies, the literature and our findings agree that KD may reduce the phyla Bacteriodetes and Actinobacteria and increase the phyla Firmicutes. Recently, the role and possible therapeutic uses of Actinobacteria, especially in gastrointestinal and systemic diseases, have been discussed [55]. Moreover, at this point, the change in the Firmicutes/Bacteriodetes ratio, which is effective in maintaining normal intestinal homeostasis and is associated with various health outcomes, is worth noting [56]. The literature presents different findings on this issue. Some studies have reported that KD decreases [57] the Firmicutes/Bacteriodetes ratio, whereas other studies have reported that it increases it [58,59]. In the present study, we observed that the Firmicutes/Bacteriodetes ratio significantly increased following the KD. Considering the effects of both Actinobacteria and this ratio, especially on metabolic health, the findings of our study suggest that KD can cause dysbiosis in the microbiota in healthy populations.

Investigating the effects of KD at the genus level showed decreased abundance of *Prevotella* and *Bifidobacter*. Previous studies investigating the effects of very low-carbohydrate diets on the microbiota have shown decreased abundance of butyrate-producing bacteria, including *Rosebruia*, *Eubacterium rectale*, and *Bifidobacterium* species [44,60]. In this direction, considering that *Bifidobacterium* species are involved in SCFA production [61], the decrease in fecal SCFA production due to KD may be explained by the decrease in *Bifidobacterium* species. Prevotella is frequently associated with fiber consumption [62]. Thus, the decrease in Prevotella may be explained by the decreased fiber consumption associated with KD. Moreover, we observed an increase in the abundance of *Oscilibacter*, *Blautia*, and *Akkermensia*. Lam et al., in their study on the effects of a high-fat diet rich in saturated fatty acids in mice, showed that a high-fat diet increased the relative abundance of *Oscilibacter* and that the increased *Oscilibacter* abundance was associated with a decreased synthesis of tight junction proteins. In our study, the increase in serum ZO levels and *Oscilibacter* abundance supported the findings of Lam et al. [63]. Furthermore, previous studies on mice have reported that KD increases *Akkermensia* abundance [54,64]. Although this increase and weight loss following KD may be related to significant decreases in blood glucose, HOMA-IR, and HbA1C values, conversely, the decrease in the relative abundance of fiber-degrading bacteria due to the decreased dietary fiber intake associated with KD and taxonomic changes in other members of the microbiota may have caused an increase in the abundance of mucus-degrading *Akkermansia*. In addition to all these, our findings showed that KD increased some genera of pathogens, including *Escherichia coli* (*E. coli*), *Klebsiella*, and *Listeria* in healthy populations. This finding was similar to the study by Lindefeldt et al. [65] who reported that KD administered to children with severe epilepsy increased *E. coli* levels. This finding suggests that KD can adversely affect healthy populations. Considering all these findings, in addition to the positive effects of KD, including weight loss and decreased insulin, HOMA-IR, and HbA1C levels, its negative effects, especially on microbiota and intestinal permeability, should not be ignored. However, it should be kept in mind that even though all participants followed a ketogenic diet, their food choices, and cooking and preparation techniques, may differ and these factors may affect the gut microbiota, albeit minimally.

### Limitations

This study has some limitations. The first is that it was conducted with a limited number of participants. Secondly, since the study population consists of only female participants, the generalizability of the findings to the society is limited. Third, since the BMI of the participants included in the study was between 25–35 kg/m^2^, it is possible that different findings may be obtained for individuals with BMI > 35 kg/m^2^. Fourth, the food groups chosen by the participants while following the ketogenic diet, the food preparation and cooking techniques, and the spices they use are also factors that can affect the microbiota. The fact that these factors could not be controlled is another limitation of this study.

## 5. Conclusions

Significant alterations in microbial composition led by ketogenic diet in healthy individuals can result in dysbiosis and long-term negative effects, including changes in fecal SCFA and serum zonulin levels. In this direction, randomized controlled studies with more participants are needed.

## Figures and Tables

**Figure 1 nutrients-16-04374-f001:**
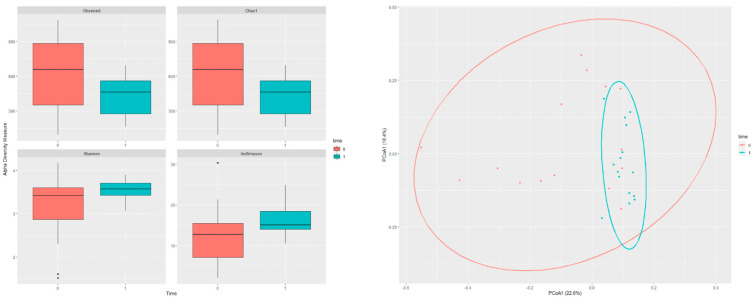
(**a**) Alpha diversity of participants before and after KD; (**b**) Beta diversity of participants before and after KD.

**Figure 2 nutrients-16-04374-f002:**
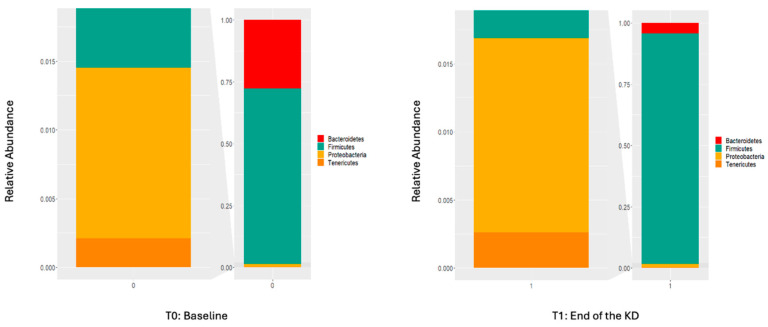
Relative abundance of the four most abundant phyla at the beginning and end of the KD.

**Figure 3 nutrients-16-04374-f003:**
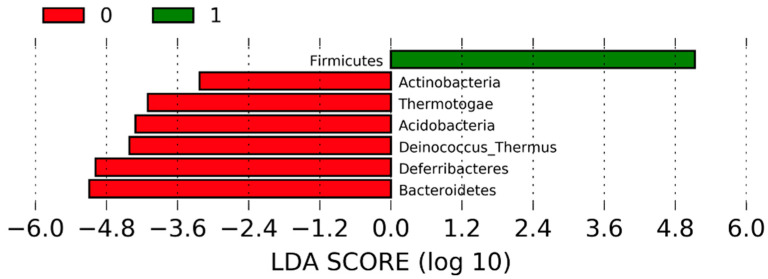
LDA scores at phylum level.

**Figure 4 nutrients-16-04374-f004:**
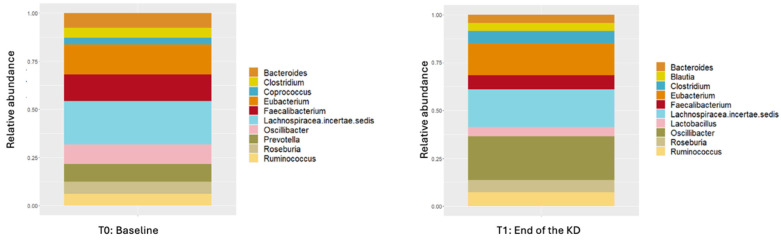
Relative abundance of the 10 most abundant genera at the beginning and end of the KD.

**Figure 5 nutrients-16-04374-f005:**
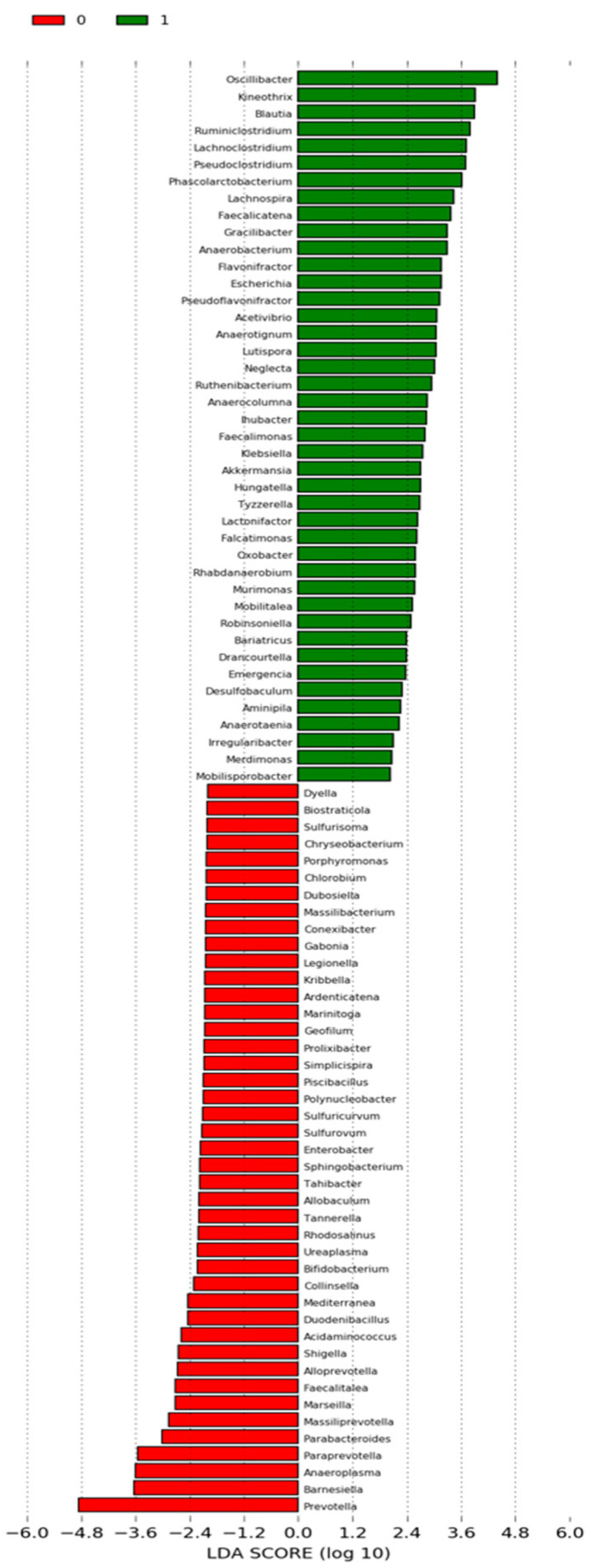
LDA scores at genus level.

**Table 1 nutrients-16-04374-t001:** Energy and nutrient intakes at baseline and after the KD.

Characteristics	T0 (n = 15)	T1 (n = 15)	*p* Value
Energy (kcal)	1624.04 ± 433.09	1516.13 ± 177.18	0.316
Carbohydrate (%)	37.73 ± 7.56	5.4 ± 0.83	0.001 *
Fiber (g)	18.33 ± 7.11	12.77 ± 2.86	0.005 *
Protein (%)	16.87 ± 2.85	22 ± 1.41	0.000 *
Fat (%)	43.73 ± 8.24	72.87 ± 1.77	0.001 *
SFA (g)	28.99 ± 9.1	35.82 ± 10.17	0.023 *
MUFA (g)	29.97 ± 11,63	61.9 ± 8.88	0.000 *
PUFA (g)	15.06 ± 6.51	18.84 ± 7.75	0.104
Cholesterol	280.65 ± 144.1	493.79 ± 81.78	0.000 *

All parameters are evaluated at baseline and after 6 weeks (T0 and T1, respectively) of the KD. KD, ketogenic diet; SFA, saturated fatty acid; MUFA, monounsaturated fatty acid; PUFA, polyunsaturated fatty acid; * *p* < 0.05.

**Table 2 nutrients-16-04374-t002:** Anthropometric measurements and body composition of the participants at baseline and after the KD.

Characteristics	T0 (n = 15)	T1 (n = 15)	*p* Value
Weight (kg)	81.75 ± 16.75	74.39 ± 15.09	0.000 *
BMI (kg/m^2^)	29.81 ± 4.74	27.12 ± 4.23	0.000 *
WC (cm)	101.47 ± 12.89	93.53 ± 12.01	0.000 *
HC (cm)	114.07 ± 11.9	108.8 ± 11.1	0.000 *
WHR	0.89 ± 0.04	0.86 ± 0.04	0.000 *
WHtR	0.61 ± 0.07	0.57 ± 0.06	0.000 *
FM (%)	36.15 ± 5.84	33.81 ± 5.88	0.000 *
FFM (%)	63.93 ± 5.99	66.19 ± 5.86	0.000 *
Water (%)	45.73 ± 4.36	47.38 ± 4.15	0.000 *

All parameters are evaluated at baseline and after 6 weeks (T0 and T1, respectively) of the KD. BMI, body mass index; FM, fat mass; FFM, fat-free mass; HC, hip circumference; WC, waist circumference; WHR, waist-to-hip ratio; WHtR, waist-to-height ratio; * *p* < 0.05.

**Table 3 nutrients-16-04374-t003:** Biochemical parameters of the participants at baseline and after the KD.

Biochemical Parameters	T0 (n = 15)	T1 (n = 15)	*p* Value
FPG	93 ± 7.49	80.47 ± 4.91	0.000 *
HbA1c	5.37 ± 0.51	5.06 ± 0.32	0.001 *
Insulin	7.86 ± 4.68	4.2 ± 1.14	0.005 *
HOMA-IR	1.86 ± 1.24	0.84 ± 0.24	0.003 *
TC	217.33 ± 39.4	214.07 ± 60.23	0.589
HDL-C	55.07 ± 10.65	53.67 ± 13.78	0.490
LDL-C	146.53 ± 30.23	151.33 ± 46.36	0.410
TG	99.2 ± 42.58	78.67 ± 21.82	0.156
LDL/HDL	2.72 ± 0.59	2.87 ± 0.81	0.211
TC/HDL	4.02 ± 0.74	4.05 ± 0.93	0.909
Urea	26.89 ± 5.96	21.83 ± 6.04	0.020 *
Uric acid	4.71 ± 0.88	5.98 ± 0.98	0.000 *
Creatinine	0.64 ± 0.09	0.54 ± 0.07	0.000 *
ALT	16 ± 4.5	15.2 ± 4.71	0.621
AST	18.2 ± 2.54	18.2 ± 4.54	0.694
CRP	4.97 ± 5.72	4.5 ± 4.92	0.650
Ca	9.44 ± 0.29	9.44 ± 0.36	1.000
K	4.51 ± 0.46	4.46 ± 0.51	0.640
ZO	27.09 ± 12.28	45.43 ± 15.59	0.001 *

All parameters are evaluated at baseline and after 6 weeks (T0 and T1, respectively) of the KD. FPG, fasting plasma glucose; HbA1c, hemoglobin A1c; TC, total cholesterol; HDL-C, high-density lipoprotein cholesterol; LDL-C, low-density lipoprotein cholesterol; TG, triglyceride; ALT, alanine transaminase; AST, aspartate transaminase; Ca, calcium; K, potassium; ZO, zonulin; * *p* < 0.05.

**Table 4 nutrients-16-04374-t004:** Fecal SCFA levels of the participants at baseline and after the KD.

Fecal SCFA	T0 (n = 15)	T1 (n = 15)	*p* Value
Acetate	373.37 ± 236.72	161.39 ± 102.07	0.006 *
Propionate	72.32 ± 44.82	20.66 ± 13.74	0.002 *
Butyrate	60.86 ± 55.09	14.86 ± 11.03	0.003 *
Total SCFA	534.21 ± 337.68	212.59 ± 128.8	0.005 *

SCFA, short-chain fatty acid; KD, ketogenic diet; * *p* < 0.05.

## Data Availability

Data will be made available on request.

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
