# Peer review of "Effects of the Ketogenic Diet on Microbiota Composition and Short-Chain Fatty Acids in Women with Overweight/Obesity"

_nutrients, 2024, doi:10.3390/nu16244374_

Round 1

Reviewer 1 Report

Comments and Suggestions for Authors

Authors performed a study about the role of ketogenic diet on gut microbiota and other biological markers. In itself, the study is not very original, as many similar papers have been published in recent years, with larger sample size; however, the authors do not take advantage of the opportunity to pick up where their predecessors left off, despite the long period between data collection (2019) and publication. Overall, these data, which the authors acknowledge are limited by the small sample size, also add to the confusion caused by the inconsistency of results that such studies have produced over the years.

This reviewer points out some errors in the text.

1. the title is wrong because the study was not conducted on a population of healthy adults, but on adult woman with overweight/obesity.

2. It is not specified how the patients' state of ketosis was assessed.

3. The microbiota, zonulin and uric acid results are consistent with some literature data, but not with others. Thus, it cannot be assumed that what was observed depends on the ketogenic diet per se (i.e., on the distribution of macronutrients and the duration of ketosis), but could more likely be an influence exerted by the choice of food (e.g., vegetable or animal protein, quantity and quality of fiber, etc.) and the spices that season it, whose role is well known both on the intestinal microbiota and on the intestinal inflammatory state.

In light of the above, it would be necessary to revise the Discussion section in depth. This would include expanding the Study Limitations section.

Reviewer 2 Report

Comments and Suggestions for Authors

An interesting study of the effect of a ketogenic diet on the metabolic parameters of the participants and the change in gut biota and permeability that resulted from the diet. The biochemical investigations were well done and the classification of gut biota changes was comprehensive.  There were obvious improvement in metabolic changes such as blood glucose, HbA1c, insulin and liver enzymes. It is a puzzle as to why the cholesterol intake increased so much on the diet (almost double) as fruit and fish were a significant part of the diet, and protein intake was limited to 1g/Kg.  Did the diet reflect an increase in red meat intake over the pre study diet and could this account for the increased cholesterol? Olive oil supplementation should not increase cholesterol, so what was the source of the increased intake? Can the authors provide an explanation for the increase ?

The dietary list in the supplemental information indicates a significant amount of leafy vegetables (Fibre) was being consumed but the SCFA fell? Did the authors expect this and can they explain why this happened ? Had the participants not stuck to the diet or was the diet a significant change in their normal intake pattern?

Had the amount of fibre in the natural diet been higher and was the ketogenic diet a decrease in fibre? The increase in gut permeability may suggest that that was the case as fibre is an important dietary component to maintain gut impermeability. The increased gut permeability would account for the increase in uric acid as a marker of an inflammatory process, most likely from the increased permeability and the leaking of lipopolysaccharide into the blood activating the innate immune mechanisms? .  What was the source of inflammation if the intake of fibre was adequate ? A decrease on the gut bacteria responsible for short chain fatty acid production was also found in the study. This was again indicting that the ketogenic diet was not providing the same amount of fibre as the natural diet of the participants? Was this the case ? If not can the authors provide an explanation for the changes in light of the dietary list in the supplements? Were the subjects actually consuming the suggested fibre intake ?

Round 2

Reviewer 1 Report

Comments and Suggestions for Authors

Thanks for the revision of your paper. It seems appropriate, now.